# Granulomatous Inflammation and Pericarditis Induced by Silk Granuloma Related to Previous Surgical Ligation of Patent Ductus Arteriosus in a Dog

**DOI:** 10.3390/vetsci9120694

**Published:** 2022-12-14

**Authors:** Chang-Hwan Moon, Woo-Jin Kim, Won-Jong Lee, Kyung-Min Kim, Hae-Beom Lee, Seong-Mok Jeong, Dae-Hyun Kim

**Affiliations:** Department of Veterinary Surgery, College of Veterinary Medicine, Chungnam National University, 99, Daehak-ro, Yuseong-gu, DaeJeon 34134, Republic of Korea

**Keywords:** patent ductus arteriosus, silk granuloma, granulomatous inflammation, pericarditis, pericardial effusion

## Abstract

**Simple Summary:**

The present report describes granulomatous inflammation and pericarditis induced by silk granuloma related to a previous surgical ligation of patent ductus arteriosus in a dog. Surgical ligation of patent ductus arteriosus is usually accomplished by tying two pliable, heavy, non-absorbable suture ligatures using materials such as silk. In veterinary medicine, silk granuloma formation associated with patent ductus arteriosus closure has been rarely reported. To our knowledge, this is the first case report describing the clinical symptoms caused by granulomatous inflammation related to the surgical ligation of a patent ductus arteriosus in a dog.

**Abstract:**

Surgical ligation of a patent ductus arteriosus is regarded as a standard treatment approach with a low complication rate if performed by experienced surgeons, and it has been performed successfully for decades in dogs. However, there are no reports describing the clinical symptoms related to granulomatous inflammation after the surgical ligation of the patent ductus arteriosus. This report describes the clinical symptoms caused by granulomatous inflammation and subacute pericarditis in a dog that had undergone the surgical closure of a patent ductus arteriosus 2 years previously. Exploratory thoracotomy was performed for subtotal pericardiectomy, and a biopsy was performed to obtain specimens for histopathological examination and culture tests. The clinical symptoms were relieved after surgery. The persistent leukocytosis improved with steroid administration. This case illustrates that the granulomatous inflammatory response associated with silk suture granuloma is a rare postoperative complication of ductal ligation. In such cases, pericardiectomy can help relieve the clinical symptoms.

## 1. Introduction

Patent ductus arteriosus (PDA) is the most common congenital cardiovascular malformation in dogs, accounting for about 25% of all congenital heart defects [1,2,3]. Generally, in dogs with PDA, the left-to-right shunt flow from the descending aorta to the main pulmonary artery (MPA) causes pulmonary overcirculation, resulting in volume overload and the dilatation of the left atrium and ventricle. This leads to pulmonary hypertension, mitral regurgitation and deterioration of the myocardium; consequently, left-sided congestive heart failure can occur [4]. Surgical treatment of PDA is recommended as soon as possible, except in some cases, such as those involving right-to-left or bidirectional shunts, because the majority of dogs with untreated PDA have a poor long-term prognosis [4,5,6]. Most dogs with PDA could die due to progressive heart failure if not treated appropriately [5]. If the disease progresses further, a bidirectional or reverse (right-to-left) shunt flow could develop as a result of chronic pulmonary overcirculation and severe pulmonary hypertension. In such cases, surgical treatment of the PDA is contraindicated. Thus, PDA closure should be performed as soon as general anesthesia can be administered to the patient.

Treatment options for PDA closure include transcatheter intervention and surgical ligation [7,8]. Transcatheter intervention has been reported as a minimally invasive procedure to close the PDA in several veterinary studies; however, it can be performed only in some patients, depending on the patient size or patent ductal morphology type. In addition, transcatheter intervention incurs additional costs, owing to the need for consumables, and the duration of the procedure and anesthesia is longer than that of surgical ligation [9]. Surgical ligation is a widely used treatment in veterinary medicine. Surgical ligation of the PDA is usually accomplished by a left fourth intercostal thoracotomy, careful dissection around the ductus arteriosus, and tying two pliable, heavy, non-absorbable suture ligatures using suture materials such as silk. The most commonly reported complications of PDA ligation are severe hemorrhage and residual shunt flow due to inaccurate ligation.

To the best of our knowledge, reports describing pericardial effusion and granuloma formation as complications after PDA ligation are limited. This is the first case report of a dog presenting clinical signs associated with pericarditis, pericardial effusion and chronic inflammation caused by a silk granuloma that occurred after PDA ligation.

## 2. Case Presentation

A 4-year-old castrated male Maltese dog weighing 3.1 kg was referred to the veterinary medical teaching hospital of Chungnam National University for leukocytosis and a high C-reactive protein level that was unresponsive to a month of antibiotic therapy. The patient presented with lethargy, anorexia, nausea, coughing, and weight loss for 2 months. The patient had undergone PDA ligation 2 years previously. The vital signs were normal, and a physical examination revealed no specific findings. Blood analysis for complete blood cell count and biochemistry showed leukocytosis (24.82 × 109 /L; range, 5.2–13.9 × 109 /L) and a severely elevated canine C-reactive protein level (123.87 mg/L; range, 0–10 mg/L).

Thoracic radiography revealed cardiomegaly (VHS, 12.0v) and a bulging MPA region. An irregular and double-layered cardiac silhouette at the 3–10 o’clock position of the heart was identified in the ventrodorsal view. Transthoracic echocardiography revealed a 25 × 14 mm hypoechoic mass at the bifurcation of the MPA between the descending aorta and the MPA (Figure 1A). The left and right ventricles and the left pulmonary artery (LPA) were compressed and deviated by the mass (Figure 1B). The peak systolic velocity of the MPA at the pulmonic valve level was approximately 0.9 m/s, which was in the normal range (range; 0.81–1.18 m/s), while the peak velocity of the turbulent flow in the LPA was as high as 2.7 m/s. Additionally, a small amount of pericardial effusion was observed around the heart (Figure 1C). Mild mitral regurgitation was observed. The E wave, A wave and E/A ratio were measured as 67 cm/s, 49 cm/s, and 1.37, respectively. Peak aortic valve velocity (AV Vmax) was 1.4 m/s, and the left ventricular outflow tract velocity time integral (LVOT VTI) was 9.3 cm. Fractional shortening (FS) was 22.39%. Residual shunt flow was not observed. A heterogeneous lesion in the anterior heart base between the aorta and the MPA (Figure 1D), including a 20 × 15 × 10 mm hypodense mass without contrast enhancement, was identified on computed tomography (CT). The LPA was narrowed compared to the right pulmonary artery due to compression by the heart base mass (Figure 1E). The thickness of the pericardium before contrast was approximately 2 mm with soft tissue density, and contrast enhancement was confirmed for the thickened pericardium after an injection of contrast media (Figure 1F). Granuloma formed by the suture material, granulomatous inflammation, heart base tumor, pericardial effusion, pericarditis, and pericardial abscess were considered in the differential diagnosis for the periaortic mass. The clinical signs were presumed to be caused by the mass and inflammation around the heart, and exploratory thoracotomy and pericardiectomy were performed to relieve the clinical symptoms and obtain biopsy specimens of the mass.

The dog was preoxygenated with 100% oxygen. Preanesthetic medications included maropitant (1 mg/kg, SC), cefazolin sodium (22 mg/kg, IV) and midazolam (0.2 mg/kg,). Propofol (1 mg/kg, IV) and ketamine (1 mg/kg, IV) were slowly injected, followed by induction with ketopol (propofol–ketamine mixture containing propofol 3 mg/kg IV and ketamine 3 mg/kg IV). Anesthesia was maintained with isoflurane, and a constant-rate infusion of remifentanil (0.1–0.3 ug/kg/min) was used for analgesia. During surgery, the dog was placed in right lateral recumbency, and the forelimbs were secured in an extended position to prevent limb movement associated with spasms of the extrinsic musculature. After the left fourth intercostal thoracotomy, the left lung lobe was confirmed to have adhered to the thoracic wall because of the previous surgery. It was carefully dissected using electrocautery and a cotton swab to expose the base of the heart. The left lung was retracted caudally using wet gauze. It was difficult to clearly identify the anatomic structures due to the severe inflammatory reactions and fibrotic changes around the heart. Thickened pericardium and a small amount of whitish pericardial effusion were identified (Figure 2A). The pericardial fluid was submitted for cytological and culture tests. A firm mass with obscure boundaries suspected to be granulomatous tissue at the site of the surgical ligation of the PDA was observed at the location between the descending aorta and the MPA (Figure 2B). Since the mass was severely adhered to the adjacent great vessels, complete resection was considered impossible, and a biopsy was performed for histopathologic examination. Subsequently, subtotal pericardiectomy was performed. After completing the intrathoracic procedure, the pleural cavity was lavaged with warm saline and inspected for air leakage or additional hemorrhage. A thoracostomy tube was placed intercostally, and the surgical site was routinely closed.

Several degenerative macrophages were observed in the sampled pericardial fluid (Figure 3), and no evidence of infection was identified in the culture test. The periaortic mass and pericardium were histopathologically diagnosed as showing granulomatous inflammation and subacute pericarditis, respectively (Figure 4). The histopathological findings were suggestive of granulation tissue with persistent chronic active inflammation from the previous surgical ligation of the PDA. A culture test was also conducted on the biopsy sample, but no evidence of infection was observed.

The preoperative clinical signs improved the day after the surgery. Since there was little thoracostomy tube drainage and no complications associated with the tube were observed, the tube was removed the day after surgery. The patient showed no specific postoperative complications except for persistent leukocytosis. The C-reactive protein levels gradually declined after surgery. On the basis of the postoperative laboratory examinations, the persistent leukocytosis was considered to be a chronic noninfectious inflammatory response; thus, prednisolone (0.5 mg/kg, PO, twice daily) was prescribed for anti-inflammatory treatment the day after surgery. The patient was in good condition and therefore received outpatient treatment after discharge. Postoperative follow-up assessments showed a slow improvement of the persistent leukocytosis after prednisolone administration. Furthermore, a decreased size of the silk granuloma and an improvement in inflammatory response around the mass were observed upon transthoracic echocardiography. The FS was measured as 34.31%, which improved compared to before surgery. A postoperative CT scan was recommended for a more accurate assessment, but the patient’s owner refused additional diagnostic imaging due to financial issues. Blood analysis conducted 10 weeks after surgery revealed that the leukocytosis had almost resolved, and the C-reactive protein level was normal. The dose of prednisolone was gradually tapered, and the administration of prednisolone was discontinued 2 months postoperatively. No specific problems, such as the recurrence of clinical symptoms, were observed at the 1-year postoperative follow-up.

## 3. Discussion

Closure is strongly recommended for PDA, except in cases where a right-to-left or bidirectional shunt flow is clearly confirmed, since the long-term prognosis without treatment is generally very poor [10,11]. PDA closure in dogs is typically performed using a transcatheter intervention or surgical ligation. In veterinary medicine, open surgical ligation is usually performed to correct PDA. Operative mortality is reported to be very low when the procedure is performed by experienced surgeons, but several complications associated with surgical ligation have been described in the veterinary literature [6,12,13]. The most critical surgical complication associated with PDA ligation is a traumatic injury to vascular structures, including the ductus arteriosus, aorta, and pulmonary artery [12,13,14,15]. Severe hemorrhage is the most common cause of operative mortality [8,14,16], and this usually occurs during blunt dissection around the medial aspect of the ductus arteriosus [17]. Other reported surgical complications include intraoperative reflex bradycardia due to a sudden increase in aortic pressure (Branham sign), postoperative residual flow, recanalization, aneurysm formation, and left pulmonary stenosis [4,8,18,19,20,21]. However, no previous reports have described postoperative pericarditis or pericardial effusion after the surgical ligation of the PDA in dogs. To our knowledge, this is the first canine case report describing the clinical symptoms caused by granulomatous inflammation and pericarditis related to the surgical ligation of the PDA in veterinary medicine.

The occurrence of suture granulomas at the site of silk sutures was reported in both human and veterinary medicine [22,23,24,25]. Granulomas can form as a reaction to infection, inflammation, autoimmune disorders, irritants, and foreign bodies [26]. Although granulomas are not inherently cancerous, granulomatous inflammation represents a form of chronic inflammatory response [27]. Silk is an organic non-absorbable, braided, multifilament suture material [28] that is frequently used in cardiovascular procedures for the ligation of large vessels because of its excellent handling characteristics and good knot security [4]. Moreover, chronic and progressive vascular occlusion is possible with relatively high tissue reactivity [4]. Infection or allergenic reactions can be more caused by using silk suture materials that are non-absorbable and multifilament, compared to when monofilament sutures are used [23]. A previous human medicine study on tissue reactions to sutures reported that silk, among several non-absorbable suture materials, is most likely to cause granuloma formation [29]. In veterinary medicine, a previous study about postoperative suture granuloma in 40 dogs reported that 17 cases (42.5%, 17/40 dogs) were dachshunds [30]. Unfortunately, only a few reports have described the occurrence of silk granulomas after ductal ligation. Scansen et al. reported a case in which a silk granuloma after PDA closure in a dog was presumed to be one of the causes of acquired pulmonary stenosis [19]. Another study described peripheral pulmonary artery stenosis in a cat [31]. The findings of that report suggested that granulomas induced by silk sutures used to ligate PDAs may have caused stenosis by compressing the pulmonary artery. In this case, it is thought that the suture granuloma was caused by non-absorbable and multifilament silk sutures used for previous PDA closure.

In human medicine, metal surgical clip application has been used as an alternative to suture ligation for the closure of friable and thin PDAs to minimize the amount of blunt dissection around the ductus and ameliorate the risk of hemorrhage [32,33]. Mandhan et al. reported that titanium clip application significantly shortened the surgical time in comparison with open surgical ligation without significant differences in the incidence of intraoperative and postoperative complications [34]. According to one previous veterinary study, hemostatic clip application for the closure of PDA in dogs was related to the risk of residual shunt flow and recanalization [35]. However, another recent study reported that the incidence of surgical complications, including hemorrhage, residual shunt flow, and recanalization, did not differ significantly between surgical ligation and hemostatic clip application in dogs with PDA [21]. Ozai et al. described that hemostatic clip application was safe and effective, with no specific surgical complications for PDA closure in cats [36]. On the basis of these points, hemostatic clip application for PDA closure in dogs may be possible, as in humans, if appropriately sized clips are used. In human medicine, no previous reports have described granuloma formation after titanium clip application in patients with PDA, but some cases of foreign-body-induced granuloma caused by clip application have been reported in breast and hepatic surgeries [37,38]. Therefore, further studies on the risk of granuloma formation after hemostatic clip application in veterinary medicine are necessary.

In the present case, clinical symptoms such as lethargy, anorexia, and coughing appeared 2 years after surgical ligation of the PDA. Exploratory thoracotomy was performed to exclude infectious causes and obtain a biopsy specimen of the mass, and granulomatous inflammation and subacute pericarditis were diagnosed on the basis of these assessments. Granuloma formation by the silk suture material used in the previous surgery and the chronic inflammatory response around the heart base were the main causes of the clinical signs. Subtotal pericardiectomy and steroid administration were effective in relieving the clinical signs.

## 4. Conclusions

In conclusion, although this is a single case report, it describes a rare postoperative complication associated with the surgical ligation of the PDA. In patients with suspected granulomatous inflammation at the surgical site of the PDA ligation, pericardiectomy and appropriate medical treatment could help relieve clinical symptoms.

## Figures and Tables

**Figure 1 vetsci-09-00694-f001:**
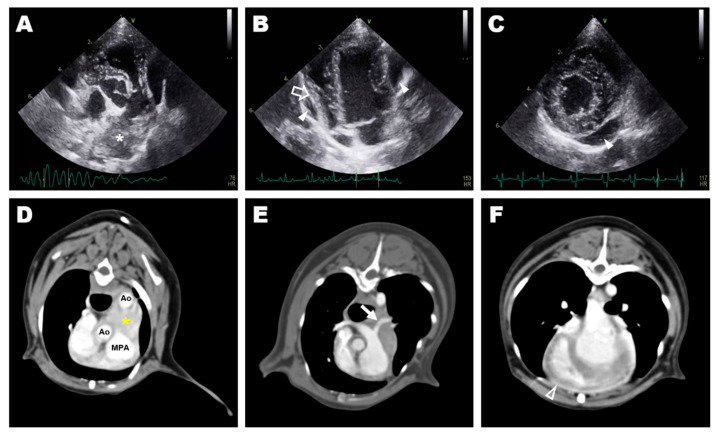
Preoperative diagnostic imaging: transthoracic echocardiography (**A**–**C**) and transverse image of computed tomography (**D**–**F**); (**A**) hyperechoic mass (asterisk) at the bifurcation of the main pulmonary artery identified at aortic root level short-axis view of the right parasternal window; (**B**) pericardial effusion (arrowhead) and deviated free wall of the right ventricle (blanked arrow) identified at the apical four-chamber view of the left parasternal window; (**C**) small amount pericardial effusion (arrowhead) identified at the short-axis view of the right parasternal window; (**D**) a heterogeneous lesion (yellow asterisk) in the anterior heart base between the aorta and the main pulmonary artery; (**E**) stenosed left pulmonary artery (arrow) caused by compression of the heart base mass; (**F**) thickened pericardium with post-contrast enhancement (blanked arrowhead).

**Figure 2 vetsci-09-00694-f002:**
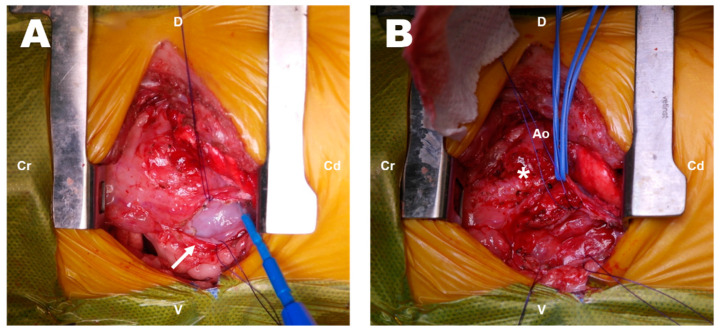
Intraoperative findings on exploratory thoracotomy (**A**,**B**). Thickened pericardium (arrow) and a lesion (asterisk) that was difficult to distinguish from the surrounding structures. Cr: cranial; Cd: caudal; D: dorsal; V: ventral; Ao: aorta.

**Figure 3 vetsci-09-00694-f003:**
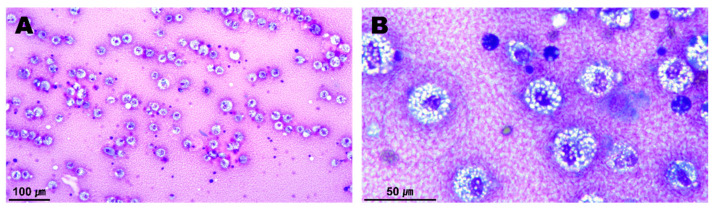
Cytology of the sampled pericardial fluid (**A**,**B**). A number of inflammatory cells appearing in the chronic inflammatory response were observed. Most of them were degenerative macrophages with vacuolated cytoplasms.

**Figure 4 vetsci-09-00694-f004:**
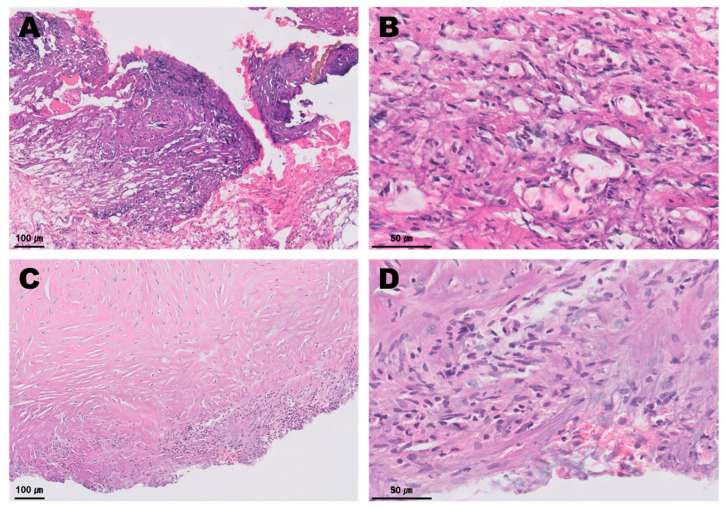
Histopathology of the periaortic mass (**A**,**B**) and the pericardium (**C**,**D**). The periaortic mass contained abundant basophilic fibrillary cell debris intermixed with bands of fibrin, reactive fibrovascular stroma containing scattered mononuclear inflammatory cells. A small amount of adipose tissue was intermixed with fibrin and loosely scattered neutrophils, macrophages, lymphocytes and plasma cells. The thickened pericardium contained loosely scattered fibrocytes, and fewer fibroblasts were observed. A small amount of basophilic fibrillar cell debris, a small number of scattered neutrophils and fewer macrophages embedded in collagen were identified.

## Data Availability

Data are available on request due to restrictions, e.g., privacy or ethical restrictions.

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
