# Peer review of "Granulomatous Inflammation and Pericarditis Induced by Silk Granuloma Related to Previous Surgical Ligation of Patent Ductus Arteriosus in a Dog"

_vetsci, 2022, doi:10.3390/vetsci9120694_

Round 1
Reviewer 1 Report
I was asked to review your case report called "Granulomatous Inflammation and Pericarditis Induced by Silk Granuloma ". The case report is interesting, and provides a useful addition to the literature given previous low numbers of cases reported in dogs with Patent Ductus Arteriosus. However, there are some concerns that have to be addressed. Some specific comments are shown below.
<Abstract>
1. I think steroid administration were most effective in this case report. Therefore, author should provide the information about steroid treatment in abstract section.
<Case presentation>
1. Granulomas is caused by an excessive immune response to suture material. Please provide the information whether the dog has a history of allergies.
2. Information on echocardiographic parameters is lacking. The author should add the information about echocardiographic parameters including heart function (For example: E wave, A wave, LVOT flow, FS, etc.), because pericarditis may result in congestive heart failure.
3. Do you use muscle-relaxing drug (Pancuronium, Pancuronium etc.)?
4. Please provide images of the surgery and explain where did you perform biopsy from?
5. When did you start prednisolone?? You should provide.
6. After prednisolone treatment, the mass size decreased?
7. Providing improved CT images will improve the quality of the paper. Would you provide the CT images after treatment?
<Discussion>
1. Please provide the discussion about the cause of granuloma in this case (dog breed? Thread type?allergy), and also discuss about risk factor (For example: You shouldn’t use silk suture in dogs with allergy, atopia, specific breed; dachshund etc.)
2. Reopen chest surgery is very risky because of the adhesion. In this case, there is a possibility that it can be improved by administering prednisolone only. Please tell me and provide the reason for performing open chest biopsy?
3. Finally, does the author recommend hemostatic clip? If you recommend, in what case do think clips should be applied? (dogs with allergy?, atopia?, specific breed?; dachshund etc.)
<Figure 1>
1. D-F are transverse images? Please provide.
Author Response
Response to Reviewers
We would like to express our gratitude to the editor and reviewers for taking the time to carefully read and to consider the report on our present study. We appreciate the opportunity to revise the manuscript by addressing the reviewers’ comments based on their constructive guidance to help you and the reviewers in your final decision. Below, we have provided detailed and concise point-by-point responses to each of the reviewer concerns in conjunction with the revised manuscript. Our responses in this document and the corresponding changes in the manuscript are written in blue.
[Reviewer: 1]
- Abstract: I think steroid administration were most effective in this case report. Therefore, author should provide the information about steroid treatment in abstract section.
Thank you for bringing this matter to our attention. We apologize for this oversight.
We have added the sentence in the abstract as follows (Line 28):
The clinical symptoms were relieved after surgery. The persistent leukocytosis improved with steroid administration. This case illustrates that the granulomatous inflammatory response associated with silk suture granuloma is a rare postoperative complication of ductal ligation.
- Case presentation - Granulomas is caused by an excessive immune response to suture material. Please provide the information whether the dog has a history of allergies.
Thank you for bringing this matter to our attention. We apologize for this oversight.
Regarding the problem you mentioned, we think as follows:
The patient in present case report had no history of allergies.
- Case presentation - Information on echocardiographic parameters is lacking. The author should add the information about echocardiographic parameters including heart function (For example: E wave, A wave, LVOT flow, FS, etc.), because pericarditis may result in congestive heart failure.
Thank you for bringing this matter to our attention. We apologize for this oversight.
We have added the word in the manuscript as reviewer’s advice (Line 83-87):
The peak systolic velocity of the MPA at the pulmonic valve level was approximately 0.9 m/s, which was in the normal range (range; 0.81-1.18 m/s), while the peak velocity of the turbulent flow in the LPA was as high as 2.7 m/s. Additionally, a small amount of pericardial effusion was observed around the heart (Figure 1C). Mild mitral regurgitation was observed. E wave, A wave and E/A ratio were measured as 67 cm/s, 49 cm/s, and 1.37, respectively. Peak aortic valve velocity (AV Vmax) was 1.4 m/s, and left ventricular outflow tract velocity time integral (LVOT VTI) was 9.3 cm. Fractional shortening (FS) was 22.39%. Residual shunt flow was not observed. A heterogeneous lesion in the anterior heart base between the aorta and the MPA (Figure 1D), including a 20 × 15 × 10 mm hypodense mass without contrast enhancement, was identified on computed tomography (CT).
- Case presentation - Do you use muscle-relaxing drug (Pancuronium, Pancuronium etc.)?
Thank you for bringing this matter to our attention. We apologize for this oversight.
Regarding the problem you mentioned, we think as follows:
No muscle-relaxing drugs was used in this case.
- Case presentation - Please provide images of the surgery and explain where did you perform biopsy from?
Thank you for bringing this matter to our attention. We apologize for this oversight.
We have added the figure and the figure legend in the manuscript as reviewer’s advice (Line 133-136):
Figure 2. Intraoperative findings on exploratory thoracotomy (A-B). Thickened pericardium (arrow), and the lesion (asterisk) that was difficult to distinguish from the surrounding structures. Cr: cranial; Cd: caudal; D: dorsal; V: ventral; Ao: aorta.
- Case presentation - When did you start prednisolone?? You should provide.
Thank you for bringing this matter to our attention. We apologize for this oversight.
We have revised the sentence in the manuscript as reviewer’s advice (Line 163):
The C-reactive protein levels gradually declined after surgery. On the basis of the postoperative laboratory examination, the persistent leukocytosis was considered to be a chronic noninfectious inflammatory response; thus, prednisolone (0.5 mg/kg, PO, twice daily) was prescribed for anti-inflammatory treatment the day after surgery. The patient was in good condition and therefore received outpatient treatment after discharge.
- Case presentation - After prednisolone treatment, the mass size decreased?
Thank you for bringing this matter to our attention. We apologize for this oversight.
We added the sentence in the manuscript as reviewer’s advice (Line 166-167):
Postoperative follow-up assessments showed a slow improvement of the persistent leukocytosis after prednisolone administration. Furthermore, decreased size of the silk granuloma and improvement of inflammatory response around the mass were observed on transthoracic echocardiography. The FS was measured as 34.31%, which improved compared to before surgery.
- Case presentation - Providing improved CT images will improve the quality of the paper. Would you provide the CT images after treatment?
Thank you for bringing this matter to our attention. We apologize for this oversight.
Regarding the problem you mentioned, we think as follows:
Unfortunately, the patient’s owner refused additional computed tomography due to financial issues.
We added the sentence in the manuscript as reviewer’s advice (Line 168-170):
The FS was measured as 34.31%, which improved compared to before surgery. Postoperative CT scan was recommended for more accurate assessment, but the patient’s owner refused additional diagnostic imaging due to financial issues. Blood analysis conducted 10 weeks after surgery revealed that the leukocytosis had almost resolved, and the C-reactive protein level was normal.
- Discussion - Please provide the discussion about the cause of granuloma in this case (dog breed? Thread type?allergy), and also discuss about risk factor (For example: You shouldn’t use silk suture in dogs with allergy, atopia, specific breed; dachshund etc.)
Thank you for bringing this matter to our attention. We apologize for this oversight.
We added the sentence in the manuscript as reviewer’s advice (Line 203-205):
Moreover, chronic and progressive vascular occlusion is possible with relatively high tissue reactivity [4]. Infection or allergenic reaction could be more caused by using silk suture materials which are non-absorbable and multifilament, compared to when monofilament suture are used [23]. A previous human medicine study on tissue reactions to sutures reported that silk, among several non-absorbable suture materials, is most likely to cause granuloma formation [29].
We added the sentence in the manuscript as reviewer’s advice (Line 207-209):
A previous human medicine study on tissue reactions to sutures reported that silk, among several non-absorbable suture materials, is most likely to cause granuloma formation [29]. In veterinary medicine, previous study about postoperative suture granuloma in 40 dogs reported that 17 cases (42.5%, 17/40 dogs) were dachshund [30]. Unfortunately, only a few reports have described the occurrence of silk granulomas after ductal ligation.
We have added the sentences in the manuscript as reviewer’s advice (Line 214-216):
The findings of that report suggested that granulomas induced by silk sutures used to li-gate PDA may have caused stenosis by compressing the pulmonary artery. In this case, it is thought that suture granuloma was caused by non-absorbable and multifilament silk sutures used for previous PDA closure.
- Discussion - Reopen chest surgery is very risky because of the adhesion. In this case, there is a possibility that it can be improved by administering prednisolone only. Please tell me and provide the reason for performing open chest biopsy?
Thank you for bringing this matter to our attention. We apologize for this oversight.
Regarding the problem you mentioned, we think as follows:
Before referred to our facility, the patient presented with lethargy, anorexia, nausea, coughing, and weight loss for 2 months. Persistent leukocytosis was unresponsive to a month of anti-biotic therapy at local hospital. The possibility that the periaortic lesion was a tumor could not be ruled out. Pericarditis was suspected in diagnostic imaging, but it was too little to collect pericardial fluid for cytological examination and culture test. Before administering steroid, there was a need to differentiate whether it was infectious or non-infectious. Since the cause of clinical symptoms was presumed to be the mass or pericarditis, exploratory thoracotomy was performed.
- Discussion - Finally, does the author recommend hemostatic clip? If you recommend, in what case do think clips should be applied? (dogs with allergy?, atopia?, specific breed?; dachshund etc.)
Thank you for bringing this matter to our attention. We apologize for this oversight.
Regarding the problem you mentioned, we think as follows:
Several literatures reported that hemostatic clip could be applicable for patent ductus arteriosus closure in dogs and cats. In human medicine, no previous reports have described granuloma formation after titanium clip application in patients with PDA, but some cases of foreign body-induced granuloma caused by clip application have been reported. Therefore, further studies on the risk of granuloma formation after hemostatic clip application in veterinary medicine are necessary. In our opinion, however, hemostatic clip, or monofilament suture material like prolene could be good alternative to silk in the cases in which there would be high risk for silk granuloma formation.
- Figure 1. - D-F are transverse images? Please provide.
Thank you for bringing this matter to our attention. We apologize for this oversight.
We have revised the figure legend in the Figure 1. as reviewer’s advice (Line 101-102):
Preoperative diagnostic imaging: transthoracic echocardiography (A-C) and transverse image of computed tomography (D-F); (A) hyperechoic mass (asterisk) at the bifurcation of the main pulmonary artery identified at aortic root level short-axis view of right parasternal window;

Reviewer 2 Report
This is a nicely-written interesting report of a rare complication following PDA ligation in a dog. The authors performed subtotal pericardiectomy to alleviate clinical signs of effusion and pericarditis. However, the rationale for pericardiectomy was not adequately presented and described using appropriate references. Complications of pericardiectomy should be also included. This discussion section should be expanded as it represents the surgical intervention to lessen the clinical signs presented. A few minor comments are listed below:
Abstract
This report describes the clinical symptoms caused.....
Introduction
Replace Patent ductus arteriosus with PDA throughout the text except in the first sentence of the introduction
Page 3 Figure 1 (E) heart base mass (arrow);
Case Presentation
Page 3 The pericardial fluid was submitted for cytology and culture.
Time of thoracostomy tube removal?
Page 4
The preoperative .........improved the day after surgery.
On the basis of the postoperative laboratory examinations, the persistent ......
Page 5 .........improvement of the persistent leukocytosis.........
...........prednisolone was discontinued 2 months
Discussion
..........significantly shortened the surgical time in comparison .......
Page 6
.......chronic inflammatory response around.........
Author Response
Response to Reviewers
We would like to express our gratitude to the editor and reviewers for taking the time to carefully read and to consider the report on our present study. We appreciate the opportunity to revise the manuscript by addressing the reviewers’ comments based on their constructive guidance to help you and the reviewers in your final decision. Below, we have provided detailed and concise point-by-point responses to each of the reviewer concerns in conjunction with the revised manuscript. Our responses in this document and the corresponding changes in the manuscript are written in blue.
[Reviewer: 2]
- Abstract: This report describes the clinical symptoms caused.....
Thank you for bringing this matter to our attention. We apologize for this oversight.
We have revised the sentence in the abstract as follows (Line 23-25):
However, there are no reports describing the clinical symptoms related to granulomatous inflammation after surgical ligation of the patent ductus arteriosus. This report describes the clinical symptoms caused by granulomatous inflammation and subacute pericarditis in a dog that had undergone surgical ligation of a patent ductus arteriosus 2 years previously. Exploratory thoracotomy was performed for subtotal pericardiectomy, and biopsy was performed to obtain specimens for histopathological examination and culture tests.
- Introduction - Replace Patent ductus arteriosus with PDA throughout the text except in the first sentence of the introduction
Thank you for bringing this matter to our attention. We apologize for this oversight.
We have revised the word in the manuscript:
The patent ductus arteriosus was replaced with PDA throughout the manuscript except for the first sentence of the introduction.
- Figure 1. - (E) heart base mass (arrow);
Thank you for bringing this matter to our attention. We apologize for this oversight.
We have revised the figure legend in the Figure 1. as reviewer’s advice (Line 108):
(D) a heterogeneous lesion (yellow asterisk) in the anterior heart base between the aorta and the main pulmonary artery; (E) stenosed left pulmonary artery (arrow) caused by compression of the heart base mass; (F) thickened pericardium with post-contrast enhancement (blanked arrow head).
- Case presentation – Page 3, The pericardial fluid was submitted for cytology and culture.
Thank you for bringing this matter to our attention. We apologize for this oversight.
We have revised the word in the manuscript as reviewer’s advice (Line 124):
Thickened pericardium and a small amount of whitish pericardial effusion were identified (Figure 2A). The pericardial fluid was submitted for cytological and culture tests. A firm mass with obscure boundaries suspected to be granulomatous tissue at the site of surgical ligation of the PDA was observed at the location of between the descending aorta and the MPA (Figure 2B).
- Case presentation - Time of thoracostomy tube removal?
Thank you for bringing this matter to our attention. We apologize for this oversight.
We have added the sentence in the manuscript as reviewer’s advice (Line 156-158):
The preoperative clinical signs improved the day after the surgery. Since there was little the thoracostomy tube drainage and no any complications associated with the tube were observed, the tube was removed the day after surgery. The patient showed no specific postoperative complications except for persistent leukocytosis.
- Case presentation - Page 4, The preoperative .........improved the day after surgery.
Thank you for bringing this matter to our attention. We apologize for this oversight.
We have revised the sentence in the manuscript as reviewer’s advice (Line 156):
The preoperative clinical signs improved the day after the surgery. Since there was little the thoracostomy tube drainage and no any complications associated with the tube were observed, the tube was removed the day after surgery.
- Case presentation - Page 4, On the basis of the postoperative laboratory examinations, the persistent ......
Thank you for bringing this matter to our attention. We apologize for this oversight.
We have revised the words in the manuscript as reviewer’s advice (Line 160-161):
The C-reactive protein levels gradually declined after surgery. On the basis of the postoperative laboratory examinations, the persistent leukocytosis was considered to be a chronic noninfectious inflammatory response; thus, prednisolone (0.5 mg/kg, PO, twice daily) was prescribed for anti-inflammatory treatment the day after surgery.
- Case presentation - Page 5, .........improvement of the persistent leukocytosis.........
Thank you for bringing this matter to our attention. We apologize for this oversight.
We have revised the word in the manuscript as reviewer’s advice (Line 165):
The patient was in good condition and therefore received outpatient treatment after discharge. Postoperative follow-up assessments showed a slow improvement of the persistent leukocytosis after prednisolone administration. Furthermore, decreased size of the silk granuloma and improvement of inflammatory response around the mass were observed on transthoracic echocardiography.
- Case presentation - Page 5, ...........prednisolone was discontinued 2 months
Thank you for bringing this matter to our attention. We apologize for this oversight.
We have revised the word in the manuscript as reviewer’s advice (Line 173):
Blood analysis conducted 10 weeks after surgery revealed that the leukocytosis had almost resolved, and the C-reactive protein level was normal. The dose of prednisolone was gradually tapered, and the administration of prednisolone was discontinued 2 months postoperatively. No specific problems, such as recurrence of clinical symptoms, were observed at the 1-year postoperative follow-up.
- Discussion - ..........significantly shortened the surgical time in comparison .......
Thank you for bringing this matter to our attention. We apologize for this oversight.
We have revised the word in the manuscript as reviewer’s advice (Line 220):
In human medicine, metal surgical clip application has been used as an alternative to suture ligation for closure of friable and thin PDA to minimize the amount of blunt dissection around the ductus and ameliorate the risk of hemorrhage [32,33]. Mandhan et al. reported that titanium clip application significantly shortened the surgical time in comparison with open surgical ligation, without significant differences in the incidence of intraoperative and postoperative complications [34]. According to one previous veterinary study, hemostatic clip application for the closure of PDA in dogs was related to the risk of residual shunt flow and recanalization [35].
- Discussion – Page 6, .......chronic inflammatory response around.........
Thank you for bringing this matter to our attention. We apologize for this oversight.
We have removed the word in the manuscript as reviewer’s advice (Line 241):
Exploratory thoracotomy was performed to exclude infectious causes and obtain a biopsy specimen of the mass, and granulomatous inflammation and subacute pericarditis were diagnosed on the basis of these assessments. Granuloma formation by the silk suture material used in the previous surgery and chronic inflammatory response around the heart base were the main causes of the clinical signs. Subtotal pericardiectomy and steroid ad-ministration were effective in relieving the clinical signs.
